# How to Diagnose Potassium Abundance and Deficiency in Tomato Leaves at the Early Cultivation Stage

**Jinxiu Song [1], Dongxian He [2,*], Jianfeng Wang [3] and Hanping Mao [1]**

[1] College of Agricultural Engineering, Jiangsu University, Xuefu Rd No.301, Jingkou District, Zhenjiang 212013, China; songjx@ujs.edu.cn (J.S.)

[2] College of Water Resources & Civil Engineering, China Agricultural University, Qinghua East Rd No.17, Haidian District, Beijing 100083, China

[3] Key Laboratory of Facility Vegetable of Jilin Province, Jilin Academy of Vegetable and Flower Sciences, Qianpeng Rd No.555, Nanguan District, Changchun 130119, China

* Correspondence: hedx@cau.edu.cn; Tel.: +86-10-62737550

**Abstract:** Potassium is one of the indispensable nutrient elements for plant growth, fruit development, and yield. The research and application of potassium nutrition diagnosis technology is the premise of scientific potassium management. However, potassium deficiency in tomato leaves, from vegetative to reproductive growth, is not easy to diagnose. To alleviate this problem, this paper proposes a suitable method of supplying potassium to tomatoes via a nutrient solution and diagnosing potassium abundance and deficiency through diagnosis methods based on ecological morphology, biological accumulation, and the photosynthetic characteristics of tomato plants. The relationship between the ecological morphology and biomass accumulation of tomatoes cultivated in the nutrient solution with potassium supply levels of 1, 4, 8, and 16 mmol/L is also discussed, and the potassium supply in the nutrient solution was studied 21 days after transplanting. The results showed that there was a significant quadratic correlation between the potassium supply in the nutrient solution and plant height and biomass accumulation, respectively. The most suitable level of potassium supply via the nutrient solution was deemed to be 10~13 mmol/L. However, if irreversible damage or severe stress to tomato plants has occurred because of potassium deficiency, there will be serious differences in the growth status of plants, and the diagnosis results will deviate greatly. In addition, the photosynthetic induction characteristics responding to the dark–light conversion of tomato leaves with potassium contents of 0.9%, 2.1%, 3.1%, and 3.3% cultivated with potassium supply amounts of 1, 4, 8, and 16 mmol/L in the nutrient solution were investigated. The results showed that tomato leaves with potassium contents of 3.1% and 3.3% had a more rapid response to dark–light conversion and higher first-order derivatives of net photosynthetic rate compared to those with potassium contents of 0.9% and 2.1%, but the first-order derivative of intercellular $CO_2$ concentration showed an opposite trend. Additionally, a quadratic correlation between leaf potassium content and $CO_2$ assimilation during 5 min of photosynthetic induction was established ($R^2 > 0.99$). According to this correlation, the suitable leaf potassium content was estimated to be 2.3~2.7%, similar to that of tomatoes cultured in the nutrient solution with a 4~8 mmol/L potassium supply. Therefore, this method can realize the rapid, non-destructive, and real-time detection of potassium content in tomato leaves based on a portable photosynthetic measurement system by establishing the relationship between leaf potassium content and net $CO_2$ assimilation during the photosynthetic induction period, therefore helping to avoid the irreversible damage caused by potassium deficiency at the later stages of plant cultivation and providing technical support for the precise fertilization of potassium in actual cultivation.

**Keywords:** photosynthetic induction; nutrient diagnosis; net $CO_2$ assimilation; nondestructive detection; light–dark conversion

## 1. Introduction

In the context of horticultural production, potassium plays an extremely important role in promoting crop photosynthesis, growth, yield, and quality [1,2]. The potassium content in plants is second only to nitrogen and plays an important role in maintaining cellular osmotic pressure, improving photosynthetic performance, promoting substance transport, and enhancing stress resistance [3,4]. Different from calcium, phosphorus, and other elements, potassium is more mobile in plants [4,5]. Potassium deficiency tends to cause plant dwarfing, leaf chlorosis, decreases in leaf area and net photosynthetic rate, and, ultimately, the inhibition of plant growth and yield [6,7]. However, the limitation of plant growth is closely related to the regulation of potassium in leaf photosynthesis because potassium can regulate stomatal movement, activate key photosynthetic enzymes, and maintain the structure of the chloroplast thylakoid membrane [8,9]. The net photosynthetic rate of leaves with potassium deficiency decreased significantly, accompanied by the down-regulation of stomatal conductivity, mesophyll conductance, and electron transport rate [10]. Although the potassium content in plants exceeds 4% of dry weight [11–13], the influences of potassium deficiency are inconspicuous and difficult to distinguish at the early stages of vegetative growth [14–16], which can adversely affect their reproductive growth at the later stages of cultivation. Conversely, plants have the capacity to carry out the "luxury absorption" of potassium, resulting in a great waste of potassium if too much potassium is applied. Therefore, effectively diagnosing the abundance and deficiency of potassium in plants at the early stage of vegetative growth is the key to promoting high yields and the quality of plants and improving potassium utilization efficiency.

The research and application of potassium nutrition diagnosis technology is the premise of scientific potassium management. Potassium diagnosis based on crop growth status and leaf symptoms is currently commonly used by most growers, and this method can help to quickly make a clear diagnosis and give general fertilization suggestions because of its intuitive, simple, and convenient characteristics [17,18]. However, potassium deficiency can also occur when the crop's appearance shows no traces of potassium deficiency. Under such circumstances, this method cannot play an active preventive role [19]. Moreover, plant growth is susceptible to environmental factors such as the climate, pests, and diseases, which can cause interference and confusion in appearance diagnosis. The traditional methods used to diagnose potassium nutrition in plants mainly include soil analysis (or nutrient solution analysis) and plant analysis (chemical analysis of plant tissue and interstitial fluid analysis) [20,21]. Soil or nutrient solution analysis can reflect the physical and chemical properties of soil and the composition of nutrient elements. However, due to many interference factors, the accuracy of the diagnostic results is relatively low [20]. Plant analysis involves determining the abundance and deficiency of nutrient elements in plants by measuring the content of nutrient elements in plant tissues and interstitial fluid. This method can directly and accurately reflect the nutritional status of horticultural crops to provide accurate nutrient information to adjust the nutrient supply scheme in time [17,21]. However, it takes a lot of time, labor and material resources from sampling to measuring results, and it is not possible to diagnose the nutrition of horticultural crops in the field. In addition, post-analysis in the laboratory requires professional analysts, many chemicals, and expensive equipment, inconveniencing real-time diagnosis. There are also some other modern diagnostic techniques, including characteristic spectral detection [22,23], ion-selective electrode detection [24], and robot vision detection [25]. Although these diagnosis methods for nutrient elements in plants have the characteristics of simplicity, rapidity, and high timeliness, they are still nascent and in need of further research.

Diagnosing potassium contents in plants involves determining the potassium concentration in the diagnostic site of the plant and establishing a regression equation through using the biomass or economic yield to determine the critical value [18,26]. However, potassium is easy to transfer and reuse in plants, so potassium deficiency in plants is a dynamic process [25]. Although there are many studies on potassium diagnosis in plants, few methods can achieve efficient, rapid, and non-destructive detection. As we all know,

potassium can influence $CO_2$ assimilation in a short time by affecting the activity of key photosynthetic enzymes, the formation of photosynthetic intermediates, and stomatal opening and closing [27–29]. Consequently, potassium significantly affects the biological accumulation and photosynthesis of plant leaves. Thus, we set out to determine whether the potassium contents of plant leaves can be diagnosed via plant photosynthetic accumulation or photosynthetic rate to provide an early indication of possible potassium deficiency in plant leaves. Few studies have focused on the diagnosis of potassium contents in plant leaves by using photosynthetic characteristics or $CO_2$ accumulation. To establish whether there is a relationship between leaf potassium content and plant biomass accumulation, photosynthetic characteristics, $CO_2$ assimilation and to achieve accurate, efficient, real-time, and non-destructive potassium diagnosis, this experiment explored the changes in growthform, biomass accumulation, photosynthetic induction characteristics, and $CO_2$ accumulation in tomato leaves caused by leaf potassium abundance and deficiency during the transition from the vegetative growth period to the reproductive growth period to provide a rapid and non-destructive method for the diagnosis of potassium abundance and deficiency in early-cultivation-stage tomato leaves.

## 2. Materials and Methods

### 2.1. Experiment Materials

This experiment was conducted in a Chinese sunlight greenhouse, and the tomato seedlings used (*Solanum lycopersicum* L. cv. *Ouguan*) were provided by Monsanto Vegetable Seed Division. The tomato seedlings were cultivated in an artificial climate chamber for one month and then transplanted in a greenhouse. In this experiment, quartz sand was used for trough cultivation with a cultivation density of 4 plants/m$^2$ and regular irrigation with steady-pressure drip irrigation belts. The nutrient solution for irrigation was adjusted to potassium concentrations of 1, 4, 8, and 16 mmol/L using $K_2SO_4$, and the missing nitrogen was supplemented by $Ca(NO_3)_2 \cdot 4H_2O$, while other nutrients remained unchanged in the Japanese garden experimental nutrient solution (Table 1). The micronutrients in the nutrient solutions were prepared according to a common formula, with pH ranging from 6.0 to 6.5. Tomatoes were grown with different levels of potassium in each of the four nutrient solutions mentioned above, and all tomato plants were cultivated in a Chinese sunlight greenhouse to ensure that all tomato plants were grown in the same environment that external environmental conditions would not affect experimental results for anything other than the potassium contents in the nutrient solutions.

**Table 1.** Fertilizer contents of nutrient solution in different potassium treatments.

| Fertilizers | K1 mg/L | K4 mg/L | K8 mg/L | K16 mg/L |
|---|---|---|---|---|
| $KNO_3$ | 101 | 404 | 808 | 808 |
| $K_2SO_4$ | 0 | 0 | 0 | 696 |
| $Ca(NO_3)_2 \cdot 4H_2O$ | 1771 | 1416 | 944 | 944 |
| $MgSO_4 \cdot 7H_2O$ | 492 | 492 | 492 | 492 |
| $NH_4H_2PO_4$ | 153 | 153 | 153 | 153 |

### 2.2. Measurement Methods

2.2.1. Plant Growth of Tomato Plants

Plant height was measured from the stem base to the terminal bud using a ruler until the tomato plant was topping. The fresh weights of the roots, stems, and leaves of tomato plants in each treatment were measured using electronic scales (YP402, Shanghai Precision Scientific Instrument Co., Ltd., Shanghai, China). After the fresh weight of each organ was weighed, it was put into a numbered kraft envelope, blanched in the oven at 105 °C for 3 h, and dried to constant weight (more than 72 h) in the oven at 80 °C. The dry weight of each organ was measured using electronic scales (FA1204B, Shanghai Precision Scientific Instrument Co., Ltd., Shanghai, China).

### 2.2.2. $CO_2$ Exchange Rate of Tomato Leaves

A portable photosynthesis measurement system (LI-6400XT, LI-COR Inc., Lincoln, NE, USA) was used as a measuring instrument. Eight tomato plants in each treatment were randomly selected, and the photosynthetic characteristics, including net photosynthetic rate, stomatal conductivity, intercellular $CO_2$ concentration, and transpiration rate of the 5th or 6th leaves from the top, were measured. The photosynthetic characteristics were measured at 9:30~11:30 a.m. on a sunny day using a leaf chamber with a red and blue light (6400-02B). The measurement parameters of the leaf chamber were set as follows: light intensity—800 μmol/m²·s, $CO_2$ concentration—500 μmol/mol, leaf chamber temperature—25 °C, and airflow rate—500 μmol/s. The continuous $CO_2$ exchange rate of tomato leaves was measured using a continuous photosynthetic measurement system on a sunny day, and the measurement conditions were consistent with the plant growth environment. The measurement method used was based on the method of Zhang et al. [30].

### 2.2.3. Photosynthetic Induction Curve

Photosynthetic induction curves of tomato leaves with different potassium contents were measured on days 21 and 22 after planting using a standard light source leaf chamber of a portable photosynthesizer. Tomato plants cultured in the four nutrient solutions were selected from the 4th or 5th matured leaf from top to bottom. Before measurement, the leaves to be measured were completely wrapped with tinfoil, and the tinfoil was removed after 2 h of dark adaptation, and the leaves were rapidly clamped into the leaf chamber for more than 5 min of dark adaptation, and the taking of measurements started following the stabilization of all photosynthetic parameters. The measurements were performed automatically, and data were recorded every 2 s. The length of measurement time was 73 min, and the light intensity was set to 0 μmol/(m²·s) from 0 to 9 min, 1000 μmol/(m²·s) from 10 to 69 min, and 0 μmol/(m²·s) from 70 to 73 min. The other measurement parameters were set as follows: leaf chamber temperature at 25 °C, airflow velocity at 500 μmol/s, $CO_2$ concentration at 500 μmol/mol, and relative humidity at 40~50%.

### 2.2.4. Net $CO_2$ Assimilation at a Given Time

The net $CO_2$ assimilation at a given time is the net $CO_2$ uptake per unit area of the leaf at a given time, which is calculated as follows:

$$\Delta CO_2 \ = \ \frac{1}{(t_2 - t_1)} \int_{t_1}^{t_2} P_n dt$$

Note: $\Delta CO_2$ is the net $CO_2$ assimilation during the photosynthetic induction period, mmol/(m² h); $P_n$ is the net photosynthetic rate, μmol/(m² s); $t_1$ and $t_2$ are the onset and termination times of photosynthetic induction, respectively, min.

### 2.2.5. Potassium Contents of Tomato Leaves

Tomato leaves were randomly selected from top to bottom. On the 21st day after transplanting, 4~5 leaves were dried, ground, and decocted using the $H_2SO_4$-$H_2O_2$ method (X20A, Shanghai Shengsheng Automatic Analytical Instruments Co., Ltd., Shanghai, China), and the resulting decoction solution was used to measure the potassium contents of the leaves using an atomic absorption emission spectrophotometer (AA-7002, Beijing East-West Analytical Instruments Co., Ltd., Beijing, China).

### *2.3. Data Statistics and Analysis*

The analysis of experimental data was completed using Microsoft Excel 2019 and SPSS 21.0 software, and Origin 9.1 software was used for graph drawing. An analysis of variance for the data was performed at the 0.05 significance level using the LSD method for multiple comparisons.

## 3. Results and Analysis

### 3.1. Diagnosis of Potassium Abundance and Deficiency Based on Plant Growth Morphology

3.1.1. Relationship between Plant Height and Potassium Supply in the Nutrient Solution

Plant growth morphology is an apparent index that reflects the effect of potassium supply in the nutrient solution on the growth and development of tomatoes. Potassium in the nutrient solution had a significant effect on plant height (Figure 1). Plant height increased first and then decreased with the increase in the potassium supply levels in the nutrient solution, especially during the flowering period and fruit set period. During the fruit set period, in the K8 and K12 treatments, the plant height of the tomato plants was significantly higher than that in the K1 and K4 treatments. However, there was no significant difference in plant height between the K8 and K12 treatments. According to the binomial fitting relationship between the potassium supply in the nutrient solution and plant height, the optimum potassium supply levels in the nutrient solution during the seedling, flowering, and fruit set periods were 10.81 mmol/L, 8.87 mmol/L, and 10.41 mmol/L, with a mean value of 10.03 mmol/L.

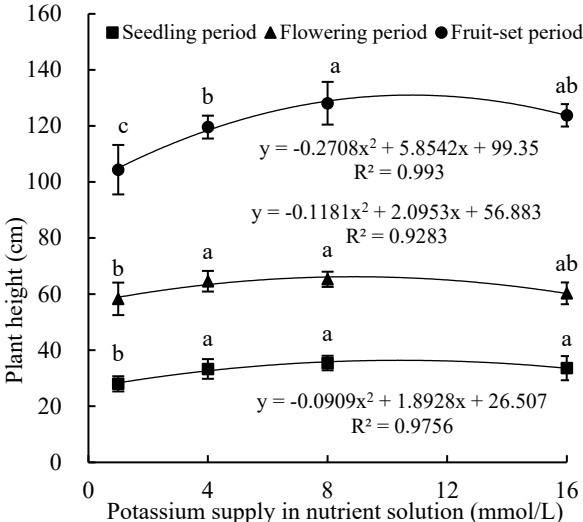

**Figure 1.** Effect of different potassium supply levels in the nutrient solution on the plant height of tomato plants during different periods. The letters above the mark indicate the results of the analysis of variance (tested using LSD method for multiple comparisons; $p < 0.05$), and different letters in the same growth period represent significant differences. The quadratic equation indicates the correlation between the potassium supply level in the nutrient solution and tomato plant height in a certain period ($n = 8$).

3.1.2. Relationship between Biomass Accumulation and Potassium Supply in the Nutrient Solution

The biomass accumulation of tomato plants mainly refers to the fresh weight and dry weight of leaves, stems, and roots. The effects of potassium supply via the nutrient solution on the biomass accumulation of the tomato plants were different (Figure 2). The biomass accumulation of tomato plants increased significantly with the increase in potassium supply levels in the nutrient solution during the seedling and flowering periods. However, there was no significant difference in the biomass accumulation of the tomato plants between the K4, K8, and K16 treatments during the seedling, flowering, and mature periods. Tomato plants mainly grow vegetatively during the seedling and flowering period, but insufficient potassium supply cannot meet the growth demand of tomato plants.

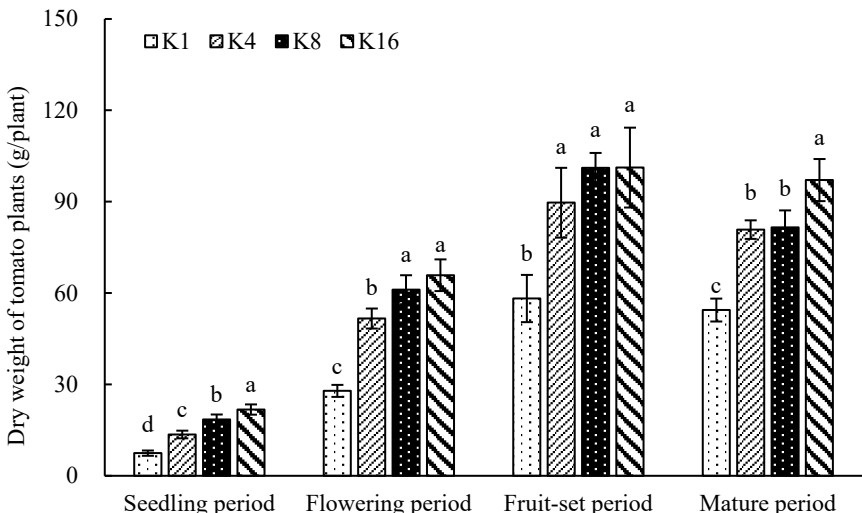

**Figure 2.** Effect of potassium supply in the nutrient solution on the dry weight of tomato plants. The letters above the histogram indicate the results of our analysis of variance (tested using the LSD method for multiple comparisons; $p < 0.05$), and different letters in same growth period represent significant differences.

There was a significant quadratic correlation between the biomass accumulation of tomato plants and the potassium supply in the nutrient solution during the different growth periods (Table 2). Based on the maximum point of the regression equation, the suitable potassium supply in the nutrient solution was inferred, which was the optimal potassium content in the nutrient solution for promoting biomass accumulation increases during the different growth periods. According to the experimental results, the most suitable potassium supply level in the nutrient solution was 11~15 mmol/L.

**Table 2.** Relationships between biomass accumulation and potassium in tomato leaves during different growth periods.

| Growth Period | Correlation Equation | $R^2$ | Maximum Value Point (mmol/L) |
|---|---|---|---|
| Seedling period | $y = -0.0798X^2 + 2.2998X + 5.3718$ | 0.9990 | 14.41 |
| Flowering period | $y = -0.3073X^2 + 7.6114X + 22.316$ | 0.9727 | 12.39 |
| Fruit-set period | $y = -0.4474X^2 + 10.286X + 50.652$ | 0.9665 | 11.50 |
| Mature period | $y = -0.1931X^2 + 5.8245X + 52.508$ | 0.8856 | 15.08 |

Note: y represents the dry weight of tomato plants, and X represents the potassium supply in the nutrient solutions.

### 3.2. Diagnosis of Potassium Abundance and Deficiency Based on the Photosynthetic Characteristics of Tomato Leaves

### 3.2.1. Photosynthetic Characteristics of Tomato Leaves

The changing trend regarding the net photosynthetic rate of the tomato leaves in each treatment was basically the same during the different growth periods (Table 3). The net photosynthetic rate of the tomato leaves, upon treatment with a high potassium supply in the nutrient solution, was significantly higher than that in treatment with a low potassium supply. The net photosynthetic rate of tomato leaves in the K1 treatment was the lowest, while there was no difference between the K8 and K16 treatments. The changing trend regarding the stomatal conductivity of the tomato leaves during the different growth periods was similar to that of the net photosynthetic rate. There was also no significant difference in stomatal conductivity between tomato leaves in the K8 and K16 treatment, except for in the flowering period. There was no significant difference in the intercellular $CO_2$ concentrations of the tomato leaves in each treatment, which may be due to the

large measurement error. The change in the transpiration rate was similar to the stomatal conductivity.

**Table 3.** Effect of potassium supply in the nutrient solution on photosynthetic characteristics of tomato leaves.

| Growth Periods | Treatments | Net Photosynthetic Rate μmol/m² s | Stomatal Conductivity mol/m² s | Intercellular $CO_2$ Concentration μmol/mol | Transpiration Rate mmol/m² s |
|---|---|---|---|---|---|
| Seedling Period | K1 | 9.0 ± 2.0 c | 0.151 ± 0.075 c | 364 ± 30 a | 3.08 ± 1.07 b |
| | K4 | 13.3 ± 3.1 b | 0.246 ± 0.061 b | 375 ± 8 a | 4.62 ± 0.84 ab |
| | K8 | 14.4 ± 1.2 ab | 0.272 ± 0.097 ab | 366 ± 36 a | 4.57 ± 1.15 ab |
| | K16 | 15.8 ± 1.2 a | 0.333 ± 0.036 a | 379 ± 13 a | 5.66 ± 0.36 a |
| Flowering Period | K1 | 11.7 ± 2.1 c | 0.233 ± 0.018 c | 389 ± 15 ab | 2.77 ± 0.14 c |
| | K4 | 13.1 ± 1.1 bc | 0.229 ± 0.049 c | 373 ± 14 b | 2.79 ± 0.41 c |
| | K8 | 13.4 ± 0.4 b | 0.295 ± 0.059 b | 392 ± 17 a | 3.45 ± 0.50 b |
| | K16 | 16.7 ± 1.0 a | 0.387 ± 0.035 a | 392 ± 8 a | 4.14 ± 0.20 a |
| Fruit-set Period | K1 | 12.8 ± 0.4 c | 0.390 ± 0.095 b | 407 ± 14 a | 6.60 ± 0.89 a |
| | K4 | 14.6 ± 1.4 b | 0.409 ± 0.077 ab | 401 ± 11 a | 6.15 ± 0.77 a |
| | K8 | 16.1 ± 0.9 a | 0.445 ± 0.043 a | 399 ± 4 a | 6.79 ± 0.66 a |
| | K16 | 15.1 ± 1.1 ab | 0.429 ± 0.081 a | 401 ± 12 a | 6.79 ± 0.79 a |
| Mature Period | K1 | 11.0 ± 3.4 c | 0.304 ± 0.046 b | 388 ± 34 a | 4.15 ± 1.8 b |
| | K4 | 14.9 ± 3.0 b | 0.371 ± 0.078 a | 377 ± 9 a | 4.88 ± 0.94 b |
| | K8 | 16.4 ± 2.1 ab | 0.376 ± 0.097 a | 371 ± 35 a | 4.85 ± 1.30 b |
| | K16 | 17.6 ± 2.2 a | 0.388 ± 0.063 a | 386 ± 12 a | 6.23 ± 0.25 a |

Note: The letters after data in the table above indicate the results of analysis of variance (tested using the LSD method for multiple comparisons; $p < 0.05$), and different letters represent significant differences in the same growth period.

### 3.2.2. Continuous $CO_2$ Exchange Rate of Tomato Leaves

Single-point photosynthetic measurement or the net photosynthetic rate of tomato leaves in a short time can only reflect the instantaneous response or the photosynthetic potential of tomato leaves in response to environmental changes. However, it cannot reflect the adaptability of tomato leaves to different potassium supply levels in the nutrient solutions for a long time. To reveal the response of tomato leaves to the change in light environment, the continuous changes in the net photosynthetic rate of the tomato leaves were monitored for a whole day. The results showed that with an increase in the potassium supply levels in the nutrient solution, the net photosynthetic rate of the tomato leaves in the K16 treatment was significantly higher than that in the other treatments, followed by the K8 treatment, while the K1 and K4 treatments were lower (Figure 3). The photosynthesis effective time also increased with the increase in potassium supply, not just the net photosynthetic rate. Potassium deficiency significantly reduced the photosynthetic effective time of the tomato leaves, and the net photosynthetic rate of the tomato leaves decreased significantly under low-light conditions. Therefore, potassium significantly reduced $CO_2$ assimilation by affecting the net photosynthetic rate and photosynthetic effective time of the tomato leaves.

Regarding the total amount of $CO_2$ absorbed for a whole day, the net $CO_2$ assimilation of the tomato leaves in each treatment increased significantly with increase potassium supply levels in the nutrient solution. It is worth noting that there was a significant linear correlation between the net $CO_2$ assimilation and the potassium supply in the nutrient solution (Figure 3), which indicates that increasing the potassium supply in the nutrient solution can not only improve the photosynthetic activity of tomato leaves but also improve the assimilation ability of tomato leaves to $CO_2$. We set out to determine whether it is possible to estimate the demand for potassium supply in the nutrient solution or diagnose the abundance and deficiency of potassium in plants through the relationship between

$CO_2$ assimilation and potassium supply. However, from the linear correlation between the potassium supply in the nutrient solution and net $CO_2$ assimilation over a whole day, it is clear that long-term net $CO_2$ assimilation cannot be used to estimate the appropriate supply of potassium in the nutrient solution.

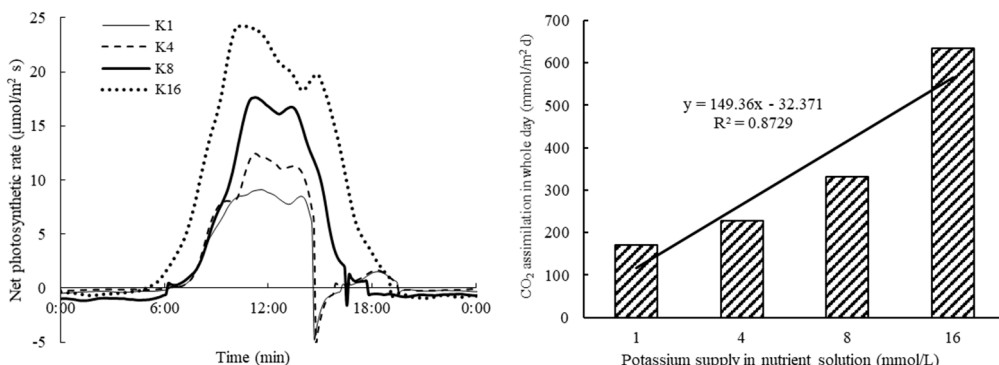

**Figure 3.** Continuous $CO_2$ exchange rate and $CO_2$ assimilation of tomato leaves cultivated in the nutrient solution with different potassium supply levels over a whole day. The continuous $CO_2$ exchange rate of the tomato leaves was measured from 6:00 to 19:00, and the time interval of each measurement was 5 min. The $CO_2$ assimilation over the whole day was the integral value of net photosynthetic rate over the whole day.

### 3.3. Diagnosis of Potassium Abundance and Deficiency Based on the Photosynthetic Induction Characteristics of Tomato Leaves

3.3.1. Photosynthetic Induction Curves of Tomato Leaves with Different Potassium Contents in Leaves

The potassium contents in the tomato leaves were detected before measuring the photosynthetic induction curves, because the effect of potassium on the photosynthetic characteristics of leaves was better explained by the potassium content in leaves than the potassium supply in the nutrient solution [31]. The results showed that the potassium contents in the tomato leaves grown in a nutrient solution with a potassium supply of 1 mmol/L reached 0.9% due to the uptake and accumulation of potassium by tomatoes during the seedling period. Tomato leaves grown in a nutrient solution supplied with 4 mmol/L of potassium reached a potassium content of 2.1%. Tomato leaves grown in the nutrient solutions supplied with 8 and 16 mmol/L of potassium did not show significant differences, reaching 3.1% and 3.3%, respectively.

The photosynthetic induction curve of tomato leaves showed a multi-stage change trend (Figure 4a). At the early lighting stage, the net photosynthetic rate showed a rapid increase in a short time (0~3 min). There were no significant differences among the leaves, except those with 0.9% potassium content, which had a lower net photosynthetic rate. The correlation between the net photosynthetic rate and potassium content in leaves reached a highly significant level with the extension of light duration (Figure 5a). The net photosynthetic rate of leaves with potassium contents of 3.1% and 3.3% increased smoothly, which was significantly higher than those of the leaves with a potassium content of 2.1%. The net photosynthetic rate of the tomato leaves tended to be stable with the continuous extension of light duration. However, the maximum net photosynthetic rate of tomato leaves with different potassium contents also showed significant differences.

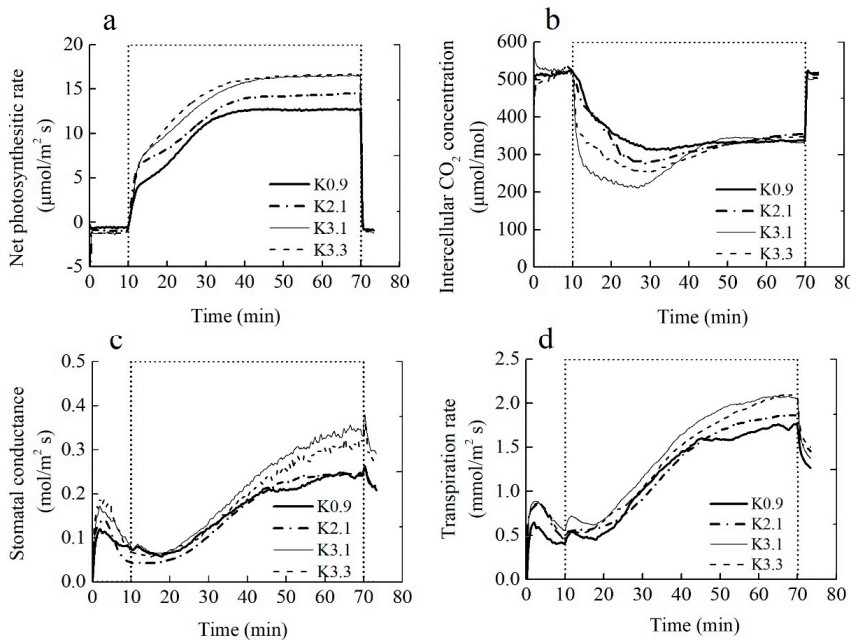

**Figure 4.** Photosynthetic induction curves of tomato leaves with different potassium contents ($n = 6$). Subfigures (**a–d**) are the photosynthetic induction curves of net photosynthetic rate, intercellular $CO_2$ concentration, stomatal conductance, and transpiration rate. The point-line box in the figures represents the lighting, and light intensity was set to 1000 $\mu$mol/m$^2$s. Data in figures were averaged every 15 points. The coarse line represents tomato leaves with a potassium content of 0.9%; the coarse dotted line represents tomato leaves with a potassium content of 2.1%. The fine line represents tomato leaves with a potassium content of 3.1%, and the fine dotted line represents tomato leaves with a potassium content of 3.3%.

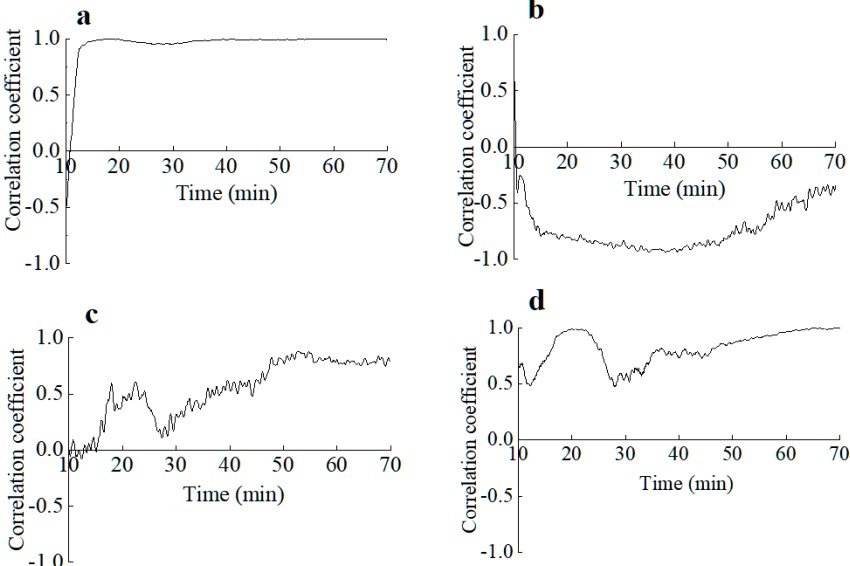

**Figure 5.** Correlations between photosynthetic characteristics and potassium content in leaves. Correlation between net photosynthetic rate (**a**), intercellular $CO_2$ concentration (**b**), stomatal conductivity (**c**), transpiration rate (**d**), and potassium content in tomato leaves.

The change in intercellular $CO_2$ concentration is the basis for analyzing the stomatal and non-stomatal limitations of leaf photosynthesis. The intercellular $CO_2$ concentration in tomato leaves showed a rapid decreasing trend within 0 to 15 min of the beginning of lighting (Figure 4b). The correlation between the intercellular $CO_2$ concentration and leaf potassium content gradually increased (Figure 5b). Among all the treatments, the

greatest decrease was observed in the leaves with a potassium content of 3.1%, and the slowest decrease was observed in the leaves with a potassium content of 0.9%. From 15 to 35 min after lighting, the intercellular $CO_2$ concentration of tomato leaves showed a slowly increasing trend, particularly in leaves with a potassium content of 3.1%. However, the difference between leaves with different potassium contents decreased gradually. With the further extension of light duration, the intercellular $CO_2$ concentration of tomato leaves showed a stable trend, and there was no significant difference in all treatments.

Stomata is the main channel for gas and water exchange between plants and external environments [31]. There was no significant change in the stomatal conductivity of the tomato leaves at the early lighting stage (0~10 min), which was different from the change in the net photosynthetic rate (Figure 4c). With the extension of light duration, the stomatal conductivity began to rise rapidly, and that of the leaves with high potassium contents was gradually greater than that of the leaves with low potassium contents. The correlation between stomatal conductivity and leaf potassium content was relatively low at the early stage but gradually increased (Figure 5c). The stomatal conductivity of leaves with potassium contents of 3.1% and 3.3% increased rapidly, while those of leaves with low potassium contents tended to stabilize when light duration was continuously extended. The photosynthetic induction curve of the transpiration rate of tomato leaves had a trend similar to that of stomatal conductivity (Figures 4d and 5d).

### 3.3.2. First-Order Derivatives of Photosynthetic Induction Curves of Tomato Leaves with Different Potassium Content

The first-order derivatives of the photosynthetic induction curves for both the stomatal conductivity and transpiration rate of the leaves in different treatments increased significantly at the early lighting stage and then stayed close to zero. The first-order derivatives started to rise above zero about 10 min after lighting (Figure 6c,d), while the first-order derivatives of the net photosynthetic rate and intercellular $CO_2$ concentration of leaves showed significant differences at the early lighting stage (Figure 6a,b). The increase rate of the net photosynthetic rate and the decrease rate of intercellular $CO_2$ concentration indicate the effect of potassium on the photosynthetic characteristics of the tomato leaves. The potassium content of the leaves with a lower increase rate in terms of net photosynthetic rate is lower, but the differences between leaves with potassium contents of 2.1%, 3.1%, and 3.3% were insignificant. The increase rate in the net photosynthetic rate of the leaves gradually decreased after 2 min of lighting, and the decrease was greater in tomato leaves with low potassium contents (Figure 6).

There were significant differences in the decrease rate in the intercellular $CO_2$ concentration, with the greatest decrease occurring in leaves with a potassium content of 3.1%, followed by leaves with a potassium content of 3.3%, and leaves with a potassium content of 0.9% experienced the smallest decrease in this regard.

### 3.3.3. Diagnosis of Potassium Abundance and Deficiency in Tomato Leaves

The net $CO_2$ assimilation in tomato leaves during the photosynthetic induction period showed the same trend as the leaf potassium content. There was no significant difference between leaves with potassium contents of 3.1% and 3.3%. The net $CO_2$ assimilation in leaves with a potassium content of 0.9% was 23.4% lower than that of leaves with a potassium content of 3.1%, and the leaves with a potassium content of 2.1% also decreased by 13.1%. In addition, there was a significant quadratic correlation between net $CO_2$ assimilation and leaf potassium content during the photosynthetic induction period, with a determination coefficient above 0.99 (Figure 7). Furthermore, there was a significant linear correlation between the potassium content of tomato leaves, net $CO_2$ assimilation during the photosynthetic induction period, and potassium supply in the nutrient solution, with a correlation coefficient of 0.99.

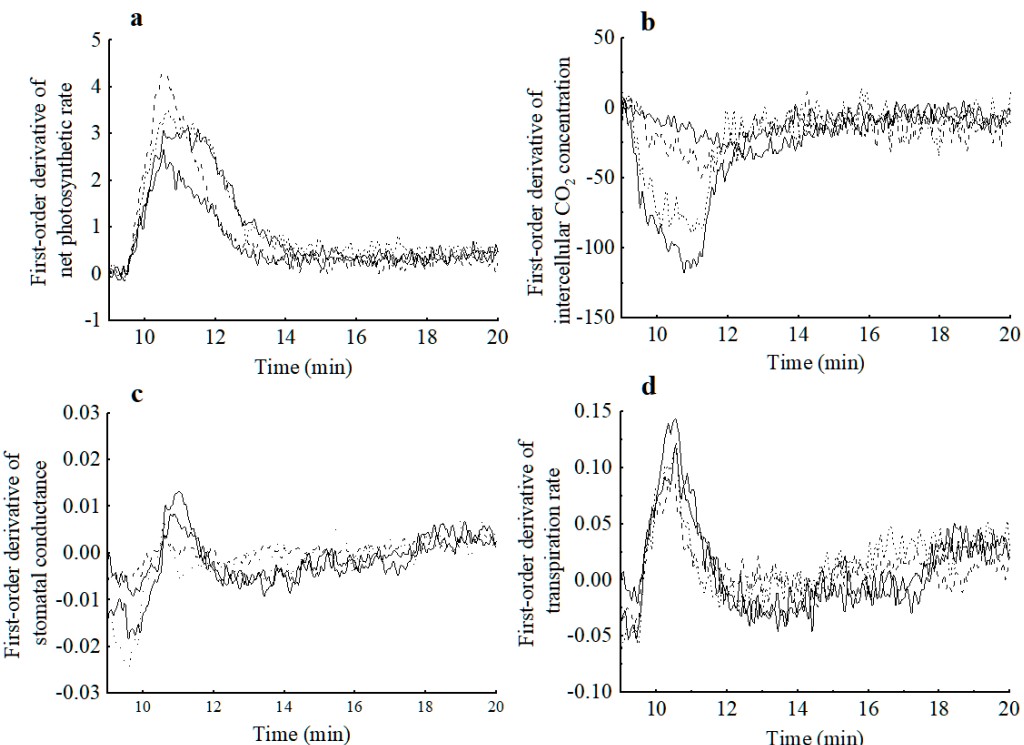

**Figure 6.** First-order derivatives of photosynthetic induction curves of tomato leaves. Subfigures (**a–d**) are the first-order derivatives of net photosynthetic rate, intercellular $CO_2$ concentration, stomatal conductance, and transpiration rate. Data in figures were averaged every 30 points. The coarse line represents tomato leaves with a potassium content of 0.9%; the coarse dotted line represents tomato leaves with a potassium content of 2.1%. The fine line represents tomato leaves with a potassium content of 3.1%; the fine dotted line represents tomato leaves with a potassium content of 3.3%.

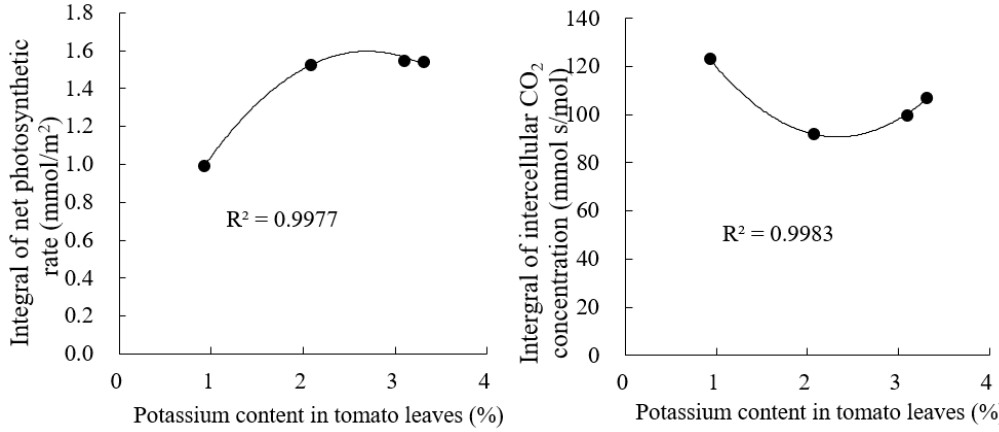

**Figure 7.** Relationships between the integral of net photosynthetic rate/intercellular $CO_2$ concentration and potassium content in tomato leaves. The integral of net photosynthetic rate is the integral of net photosynthetic rate within 5 min of the beginning of lighting in the photosynthetic induction curve, and the integral of intercellular $CO_2$ concentration is the integral of intercellular $CO_2$ concentration within 5 min of the beginning of lighting in the photosynthetic induction curve.

The significant correlation between net $CO_2$ assimilation and leaf potassium content during the photosynthetic induction period in tomato leaves can also be observed in Figure 2. However, the correlation between net photosynthetic rate and the intercellular $CO_2$ concentration of the photosynthetic induction curve in tomato leaves and leaf potassium content reached a highly significant level after 5 min of lighting. Consequently, the integrals of net photosynthetic rate and intercellular $CO_2$ level within 5 min of lighting were

selected as the characteristic variables, and it was found that they were significantly and quadratically correlated with the leaf potassium content. Suitable potassium content in tomato leaves can promote leaf photosynthesis, resulting in maximum net $CO_2$ assimilation and the lowest integral of intercellular $CO_2$ concentration. Therefore, according to the extreme points of the correlation curves of leaf potassium content with the net $CO_2$ assimilation and the integral of intercellular $CO_2$ concentration within 5 min of the beginning of lighting during the photosynthetic induction period, the suitable content of potassium in tomato leaves can be estimated. The results of this experiment show that the suitable levels of potassium in tomato leaves, estimated using net $CO_2$ assimilation and the integral of intercellular $CO_2$ concentration, approximately range between 2.3% and 2.7%, respectively.

## 4. Discussion

### 4.1. Diagnosis of Potassium Abundance and Deficiency Based on Plant Growth Morphology

Potassium, the most abundant cation in plant cells, maintains the physiological and biochemical processes of plants in various ways [32]. Severe potassium deficiency leads to insufficient potassium concentration in the intercellular substance, which affects the growth morphology and physiological metabolism of plants [33,34]. The most intuitive performance is the effect of potassium on plant growth morphology [35]. It was found in this experiment that there was a significant quadratic correlation between the plant height of tomato plants and the potassium supply in the nutrient solution, and the suitable potassium supply was found to be 10 mmol/L, slightly higher than the potassium content in the Japanese horticultural experimental nutrient formula. Therefore, plant growth morphology can be used as an effective indicator to diagnose the potassium supply levels in the nutrient solutions. However, if irreversible damage or severe stress to tomato plants has occurred because of potassium deficiency, there will be serious differences in the growth status of plants, and the diagnosis results will deviate greatly [20]. Therefore, plant growth morphology can be used to characterize severe potassium deficiency stress in plants, which reflects the effect of potassium deficiency to a certain extent [10].

Potassium deficiency will reduce the photosynthetic capacity of plant leaves, thus affecting biomass accumulation [36]. The results showed that there was a significant quadratic correlation between the biomass accumulation of tomato plants and the potassium supply in the nutrient solution during different growth periods, and the suitable potassium supply in the nutrient solution is 11~13 mmol/L, slightly higher than the diagnosis result based on plant height. However, photosynthates are more transported to vegetative organs at the early stage of vegetative growth, increasing the volume of tissue and organs. Therefore, biomass accumulation can reflect the effect of potassium on the photosynthesis of plant leaves. On the contrary, potassium content in leaves can be reflected by the change in biomass accumulation. However, more photosynthates are transported to the fruit during the reproductive growth period [6], and the potassium content in the leaves is not easy to accurately diagnose. In addition, the calculation of biomass accumulation requires the plant to be damaged; therefore, the non-destructive detection of potassium content in plant leaves cannot be realized.

### 4.2. Diagnosis of Potassium Abundance and Deficiency Based on the Photosynthetic Characteristics of Tomato Leaves

The photosynthetic characteristics of plant leaves are easy to obtain using the contemporary measurement techniques. However, $CO_2$ concentration and Rubisco activity in chloroplasts are the most important factors for determining leaf photosynthesis under sufficient light conditions. A large number of studies have focused on how to improve the transport capacity of $CO_2$ from the leaf surface to the carboxylation site of the Rubisco enzyme and the activity of the Rubisco enzyme to improve leaf photosynthesis [37]. In fact, many environmental factors or stresses can affect the photosynthesis of plant leaves. An increase in water stress will lead to the contraction of crop stomata, so the entry of $CO_2$ into the cell is blocked [38]. Additionally, 20% of the nitrogen in plants is used to synthesize the

Rubisco enzyme [39], and increasing nitrogen in leaves can reduce abscisic acid content to promote stomatal opening [40]. High-temperature stress can reduce the stomatal aperture and promote the oxidation reaction of Rubisco, which eventually leads to an increase in photorespiration and a decrease in $CO_2$ assimilation [41,42]. There have been many studies on the effects of potassium on the photosynthetic characteristics of plant leaves. The results of this experiment are consistent with those of most researchers [9,43]. What we want to express is that the potassium content of leaves will affect the $CO_2$ assimilation of leaves. The correlation between $CO_2$ assimilation and potassium supply in the nutrient solution within a certain period was used to estimate the suitable potassium content in the nutrient solution. The results of this experiment showed that the net photosynthetic rate and stomatal conductivity of leaves increased gradually with the increase in potassium supply in the nutrient solution. However, unexpectedly, there was a linear correlation between $CO_2$ assimilation and potassium content in leaves throughout the day, so it is difficult to estimate the suitable potassium supply in the nutrient solution. The main reason for this may be that the potassium in leaves not only affects the photosynthetic activity of leaves but also prolongs the photosynthetic effective time [44,45].

In addition, it is difficult to measure $CO_2$ assimilation throughout the day and detect the potassium content of leaves in real-time. The potassium supply in a nutrient solution is different from that in leaves. Because of the influence of environmental factors and other nutrient elements, the potassium content in a nutrient solution is not necessarily absorbed in equal proportion [46]. Therefore, it may be easier to quickly establish the relationship between potassium and $CO_2$ assimilation by using potassium content in leaves.

### 4.3. Diagnosis of Potassium Abundance and Deficiency Based on the Photosynthetic Induction Characteristics of Tomato Leaves

Lu et al. (2016) established the relationship between potassium deficiency in crop cytoplasm and crop growthform, photosynthetic characteristics, and dry matter accumulation, indicating that the critical reason for the inhibition of crop growth and development caused by potassium deficiency is that the potassium content in leaves is lower than its critical concentration [10]. After a relatively long period of dark acclimation, it would take some time for plant leaves to reach the maximum net photosynthetic rate under lighting, and this delayed process is called photosynthetic induction [47,48]. There is also a photosynthetic induction process in the net photosynthetic rate of plant leaves in response to light in a changing light environment [49–51]. This photosynthetic induction process is closely related to the activation of photosynthetic enzymes [52], the formation of photosynthetic intermediates [47], and light-driven stomatal opening [53]. Photosynthetic induction has the potential to be used to detect plants' adaptation to light and rapid responses to changes in other environmental conditions, such as light–dark transition [54,55], $CO_2$ concentration changes [56], temperature stress [53], and drought [57].

The photosynthetic induction of tomato leaves is a multi-stage physiological process. Potassium deficiency in plant leaves leads to delayed chloroplast development and easy disintegration, which affects the primary reaction of plant photosynthesis [58]. In the process of dark–light conversion, the $CO_2$ in intercellular substance and vacuole was consumed preferentially by plant photosynthesis due to stomatal closure [59]. With the extension of lighting, $K^+$ begins to flow inward because of the activation of the light signal, which promotes plasma membrane hyperpolarization and makes the stomata open slowly [60]. In addition, potassium can increase the number of stomata in leaves [3]. Therefore, it was found that non-stomatal limitation promoted a decrease in the net photosynthetic rate of tomato leaves at the early stage of potassium deficiency through the dynamic change in the photosynthetic induction curve, and potassium deficiency delayed the response of leaves to light, resulting in a delay in photosynthetic induction and a difference in $CO_2$ assimilation.

For this paper, the photosynthetic induction characteristics of tomato leaves with different potassium contents were analyzed. It was found that the potassium content of tomato leaves was significantly correlated with the net photosynthetic rate, stomatal

conductivity, intercellular $CO_2$ concentration, and transpiration rate during a short period of photosynthetic induction. The first-order derivative of the photosynthetic induction curve was analyzed to determine further the relationship between the potassium content of tomato leaves and the photosynthetic induction curve. It was found that there was a significant positive correlation between the change rate of net photosynthetic rate, stomatal conductivity, intercellular $CO_2$ concentration, transpiration rate, and leaf potassium content. Of course, it is difficult to estimate the potassium contents in leaves using this correlation. Therefore, a quadratic correlation between leaf potassium content and $CO_2$ assimilation during 5 min of photosynthetic induction was established, with a correlation coefficient of 0.99. According to this correlation, the suitable leaf potassium content was estimated to be 2.3~2.7%, similar to those of tomato plants cultured in the nutrient solution with 4~8 mmol/L potassium supply.

## 5. Conclusions

There are great disadvantages in the estimation of potassium supply in the nutrient solution using growth morphology, biological accumulation, and photosynthetic characteristics, and the diagnostic accuracy of leaf potassium content and $CO_2$ assimilation during the photosynthetic induction period is higher. Combined with the response of each characteristic index of photosynthetic induction during the dark–light transition and its first-order derivatives, the response of leaves with different potassium contents at the beginning of the vegetative growth period to light during photosynthetic induction differs, which in turn allows for the application of photosynthetic induction to make a timely diagnosis of early potassium abundance and deficiency in tomato leaves. By integrating the net photosynthetic rate and intercellular $CO_2$ concentration within 5 min, the optimum leaf potassium content was presumed to be 2.3~2.7%. Hence, the photosynthetic induction method at 1000 $\mu$mol/m$^2$ s can be used to rapidly detect the potassium contents of tomato leaves at the early stage of the vegetative growth period and make a diagnosis of potassium abundance and deficiency in tomato leaves. This method can realize the rapid, non-destructive, and real-time detection of potassium content in leaves through a portable photosynthetic measurement system that has good application prospects.

However, the response of the photosynthetic induction characteristics of tomato leaves to potassium abundance and deficiency was analyzed at the phenological level, and the mechanism of potassium action on photosynthetic enzyme activity and electron transport still needs to be explored further in order to determine the internal mechanism of the effect of leaf potassium on photosynthetic induction characteristics. Moreover, the diagnosis of potassium abundance and deficiency in tomato plants deserves further study in the context of severe potassium deficiency or the reproductive growth period. It is foreseeable that the combination of plant photosynthetic induction technology and plant nutrient physiology will be an important direction for future research on the diagnosis of plant element abundance and deficiency.

**Author Contributions:** Conceptualization, J.S. and D.H.; methodology, J.S. and J.W.; software, J.S. and J.W.; validation, J.S. and J.W.; formal analysis, H.M.; writing—original draft preparation, J.S. and J.W.; writing—review and editing, D.H.; project administration, D.H. and H.M.; funding acquisition, J.S. All authors have read and agreed to the published version of the manuscript.

**Funding:** This work was funded via a postdoctoral fund in Jiangsu Province (grant number: [202s0Z308]), the Key Laboratory of Modern Agricultural Equipment and Technology of Ministry of Education (grant number: [JNZ201909]), and the Priority Academic Program Development of Jiangsu Higher Education Institutions (grant number [PAPD-2018-87]).

**Data Availability Statement:** The data presented in this study are available on request from the corresponding author.

**Conflicts of Interest:** The authors declare no conflict of interest.

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
