# Peer review of "How to Diagnose Potassium Abundance and Deficiency in Tomato Leaves at the Early Cultivation Stage"

_horticulturae, doi:10.3390/horticulturae9111225_

Round 1
Reviewer 1 Report
Comments and Suggestions for Authors
The article discusses the event of potassium deficiencies in the tomato plants and possibility to measure it. The article should be improved prior the acceptance:
(1) In introduction, authors should explain why potassium is so important for plants compared to other elements.
(2) Abstract should be written more general with the implification on the society.
(3) Methods should be improved and Design of Experiments should be provided. Maybe you don't need to have all these experiments if it is well designed from the beginning.
(4) Can you please explain if there is an option to enrich potassium in soil using other biomasses "Reactivity effects of inorganic content in biomass gasification: a review"? Can you also explain the impact of other elements (Si, N, Mg) on the tomato growth and model their impact on other biomass types?
(5) Can you also add a discussion section where you present your results and talk about use of such tomato plants or waste in other industries? For example, would the K content have an impact on the pyrolysis or combustion of plants? That would make your article interesting for the industrial use.
(6) Short your conclusion and emphasize the novelty of your article.
(4)
Author Response
Dear Reviewer
Thank you very much for your comments and professional advice on our manuscript. These opinions help us to improve the quality and standardization of the manuscript. We have made extensive corrections to our previous manuscript according to your comments and suggestions, the detailed corrections are listed in the attachment. The reviewer comments are laid out below in italicized font and specific concerns have been numbered. Our response is given in normal font and changes/additions to the manuscript are given in the red text. We hope that the manuscript will be accepted for publication in horticulturae journal. If there are any other modifications we could make, we would like very much to modify them and we really appreciate your help.
Thank you again for your suggestions. Looking forward to hearing from you soon, thank you and best regards.
Yours sincerely,
Jinxiu Song, PhD, First Author
songjx@ujs.edu.cn

Reviewer 2 Report
Comments and Suggestions for Authors
The manuscript presents a new non-invasive approach to estimate the potassium content of tomato plants from dark-light photosynthetic induction characteristics. This concerns the phase of the first few minutes after light is supplied. The authors analyzed carefully how the following parameters varied with time during this phase: net photosynthetic rate, stomatal conductivity, intercellular CO2 concentration and transpiration rate. They found that the optimal leaf potassium content is 2.3-2.7%. The new approach should be very relevant for further studies that aim to optimize the parameters for plant growth with relevance to agricultural practice. The presentation of the paper is generally good. Figure 3 (right panel) seems to be missing data - the authors should rectify this.
Author Response
Dear Reviewer
Thank you for your recognition of our work, and thank you very much for your comments and suggestions on this manuscript. We have revised the manuscript according to your comments. We are very sorry for our negligence in drawing Figure 3. It may be due to the problem of image format adjustment. We have made corrections according to the Reviewer’s comments, the detailed corrections are listed in the attachment. If there are any other modifications we could make, we would like very much to modify them and we really appreciate your help.
Thank you again for your suggestions. Looking forward to hearing from you soon, thank you and best regards.
Yours sincerely,
Jinxiu Song, PhD, First Author
songjx@ujs.edu.cn

Reviewer 3 Report
Comments and Suggestions for Authors
The paper reported a series of experiments carried out with the aim of demonstrating the possibility of making an early diagnosis of potassium content in plant leaves by using photosynthetic characteristics or CO2 accumulation, measured with no-destructive methods.
However, in my opinion, this hypothesis is wrong. Photosynthesis is a complex process that can be influenced by many environmental variables and different types of stress (oxidative, nutritional, and water for example).
Therefore, the work must be strongly revised in the introductory part and the discussion/conclusions sections. In particular:
Title: I will prefer less emphasis on the possibility of using photosynthesis as a diagnostic method for deficiency of potassium
Rows 87-97. It is necessary to insert some criticism about the influence of different stress on the level of photosynthesis. It is better to insert that the aim of the work is to describe how different levels of potassium can influence the value of photosynthesis and the possibility of creating an early alert for a possible potassium deficiency must be confirmed by other parameters such as the potassium content in the leaves.
Discussion: in the three discussion sessions, a new part relating to how other stresses can modify the level of photosynthesis must also be included (this part could be bibliographic). As a matter of fact, the relationships found by the authors are important, but a diagnosis on potassium deficiency cannot be reached based only on photosynthesis
Conclusions: line 507-512. Bibliographic data do not support this conclusion because the phenomenon of reduction of photosynthesis can also be caused by other stresses and not only by a deficiency of potassium
Other corrections required:
Fig 1. The left graphic (the one with bars) reported the same information reported on the right. Please insert the statistic results present in the left graphic on the right, and then delete the left
Fig. 2. Fresh and dry matter have the same trend. Please, delete the graph with fresh biomass and use only the dry matter graph
Fig. 3: The right graph is without data. Please remove it.
Comments on the Quality of English LanguageMinor revisions are required.
Author Response

(The authors gave the same response as above.)

Round 2
Reviewer 1 Report
Comments and Suggestions for Authors
All comments were well integrated
Comments on the Quality of English LanguageNo problem with English writing.
Author Response
Dear Reviewer
Thank you very much for reviewing this manuscript carefully again and we appreciate the positive comments from the reviewer. If there are any other modifications we could make, we would like very much to modify them and we really appreciate your help. Thank you again for your suggestions.
Yours sincerely,
Jinxiu Song, PhD, First Author
songjx@ujs.edu.cn
Reviewer 3 Report
Comments and Suggestions for Authors
The authors had accepted most of the revisions.
Anyway, the paper needs to be improved, according to the following instructions:
Row 125: the phase must be rewritten since it cannot start with the conjunction and. Please there are also a typing error: coltivatesd must be “cultivated”
Rows 200-201-202. The capitation is not enough descriptive. Please insert more information about the significance of letters and the equation. The number of sample used for calculating the error bars (or the standard deviation bars?)
Row 221 Caption of figure 2: same problems of the figure 1
Row 241: please maintain the caption of the table 3 in the same page of the table. In the caption more information about the statistic letter are calculated must be inserted
Row274: in the caption the term potassium must be corrected. The two graphs must be described better
Rows 392-394. This text must be removed since the graphs must be near to caption
Comments on the Quality of English LanguagePlease pay attention, since there were some little typing errors
Author Response
Dear Reviewer
Thank you very much for your comments and professional advice on our manuscript. We are very sorry that there are still many mistakes in the manuscript. We have made careful corrections to our previous manuscript according to your comments and suggestions, the detailed corrections are listed below. If there are any other modifications we could make, we would like very much to modify them and we really appreciate your help.
Thank you again for your suggestions and best regards.
Yours sincerely,
Jinxiu Song, PhD, First Author
songjx@ujs.edu.cn
